# Towards Learning to Speak and Hear Through Multi-Agent Communication over a Continuous Acoustic Channel

## Abstract

While multi-agent reinforcement learning has been used as an effective means to study emergent communication between agents, existing work has focused almost exclusively on communication with discrete symbols. Human communication often takes place (and emerged) over a continuous acoustic channel; human infants acquire language in large part through continuous signalling with their caregivers. We therefore ask: Are we able to observe emergent language between agents with a continuous communication channel trained through reinforcement learning? And if so, what is the impact of channel characteristics on the emerging language? We propose an environment and training methodology to serve as a means to carry out an initial exploration of these questions. We use a simple messaging environment where a "speaker" agent needs to convey a concept to a "listener". The Speaker is equipped with a vocoder that maps symbols to a continuous waveform, this is passed over a lossy continuous channel, and the Listener needs to map the continuous signal to the concept. Using deep Q-learning, we show that basic compositionality emerges in the learned language representations. We find that noise is essential in the communication channel when conveying unseen concept combinations. And we show that we can ground the emergent communication by introducing a caregiver predisposed to "hearing" or "speaking" English. Finally, we describe how our platform serves as a starting point for future work that uses a combination of deep reinforcement learning and multi-agent systems to study our questions of continuous signalling in language learning and emergence.

## 1 Introduction

Reinforcement learning (RL) is increasingly being used as a tool to study language emergence (Mordatch & Abbeel, 2017; Lazaridou et al., 2018; Eccles et al., 2019; Chaabouni et al., 2020; Lazaridou & Baroni, 2020). By allowing multiple agents to communicate with each other while solving a common task, a communication protocol needs to be established. The resulting protocol can be studied to see if it adheres to properties of human language, such as compositionality (Kirby, 2001; Geffen Lan et al., 2020; Andreas, 2020; Resnick et al., 2020). The tasks and environments themselves can also be studied, to see what types of constraints are necessary for human-like language to emerge (Steels, 1997). Referential games are often used for this purpose (Kajic et al., 2020; Havrylov & Titov, 2017; Yuan et al., 2020). While these studies open up the possibility of using computational models to investigate how language emerged and how language is acquired through interaction with an environment and other agents, most RL studies consider communication using *discrete* symbols.

Spoken language instead operates and presumably emerged over a *continuous* acoustic channel. Human infants acquire their native language by being exposed to speech audio in their environments (Kuhl, 2005); by interacting and communicating with their caregivers using continuous signals, infants can observe the consequences of their communicative attempts (e.g. through parental responses) that may guide the process of language acquisition (see e.g. Howard & Messum (2014) for discussion). Continuous signalling is challenging since an agent needs to be able to deal with different acoustic environments and noise introduced by the lossy channel. These intricacies are lost when agents communicate directly with discrete symbols. This raises the question: Are we able

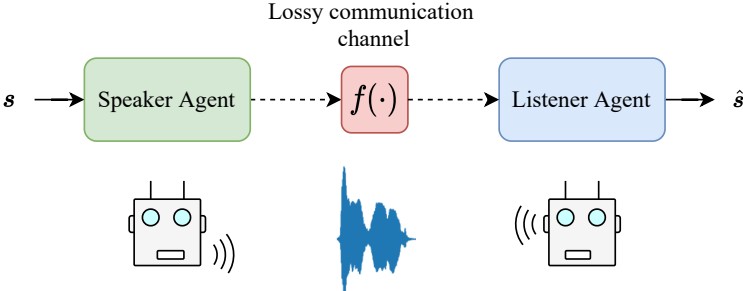

Figure 1: Environment setup showing a Speaker communicating to a Listener over a lossy acoustic communication channel $f$.

to observe emergent language between agents with a continuous communication channel, trained through RL? This paper is our first step towards answering this larger research question.

Earlier work has considered models of human language acquisition using continuous signalling between a simulated infant and caregiver (Oudeyer, 2005; Steels & Belpaeme, 2005). But these models often rely on heuristic approaches and older neural modelling techniques, making them difficult to extend; e.g. it isn't easy to directly incorporate other environmental rewards or interactions between multiple agents. More recent RL approaches would make this possible, but as noted, has mainly focused on discrete communication. Our work here tries to bridge the disconnect between recent contributions in multi-agent reinforcement learning (MARL) and earlier literature in language acquisition and modelling (Moulin-Frier & Oudeyer, 2021).

One recent exception which do use continuous signalling within a modern RL framework is the work of Gao et al. (2020). In their setup, a Student agent is exposed to a large collection of unlabelled speech audio, from which it builds up a dictionary of possible spoken words. The Student can then select segmented words from its dictionary to play back to a Teacher, which uses a trained automatic speech recognition (ASR) model to classify the words and execute a movement command in a discrete environment. The Student is then awarded for moving towards a goal position. We also propose a Student-Teacher setup, but importantly, our agents can generate their own unique audio waveforms rather than just segmenting and repeating words exactly from past observations. Moreover, in our setup an agent is not required to use a pretrained ASR system for "listening".

Concretely, we propose the environment illustrated in Figure 1, which is an extension of a referential signalling game used in several previous studies (Lewis, 1969; Lazaridou et al., 2018; Chaabouni et al., 2020; Rita et al., 2020). Here $s$ represents one out of a set of possible concepts the Speaker must communicate to a Listener agent. Taking this concept as input, the Speaker produces a waveform as output, which passes over a (potentially lossy) acoustic channel. The Listener "hears" the utterance from the speaker. Taking the waveform as input, the Speaker produces output $\hat{s}$. This output is the Listener's interpretation of the concept that the Speaker agent tried to communicate. The agents must develop a common communication protocol such that $s = \hat{s}$. This process encapsulates one of the core goals of human language: conveying meaning through communication (Dor, 2014). To train the agents, we use deep Q-learning (Mnih et al., 2013).

Our bigger goal is to explore the question of whether and how language emerges when using RL to train agents that communicate via continuous acoustic signals. Our proposed environment and training methodology serves as a means to perform such an exploration, and the goal of the paper is to showcase the capabilities of the platform. Concretely, we illustrate that a valid protocol is established between agents communicating freely, that basic compositionality emerges when agents need to communicate a combination of two concepts, that channel noise affects generalisation, and that one agent will act accordingly when the other is made to "hear" or "speak" English. At the end of the paper, we also discuss questions that can be tackled in the future using the groundwork laid here.

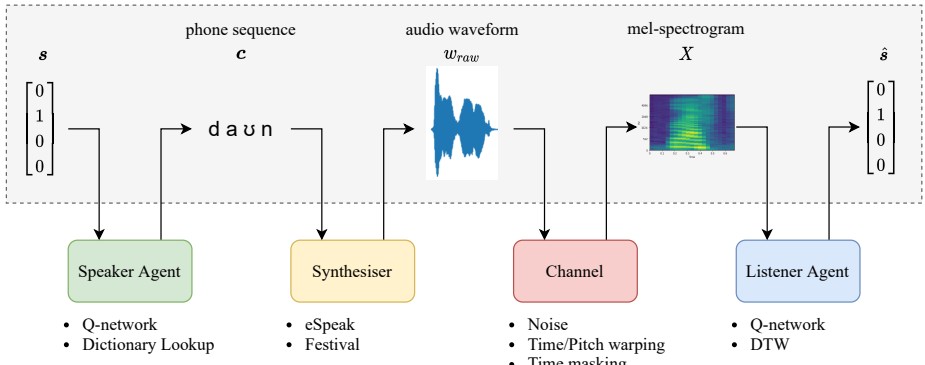

Figure 2: Example interaction of each component and the environment in a single round.

## 2 ENVIRONMENT

We base our environment on the referential signaling game from Chaabouni et al. (2020) and Rita et al. (2020)—which itself is based on Lewis (1969) and Lazaridou et al. (2018)—where a sender must convey a message to a receiver. In our case, communication takes place between a Speaker and a Listener over a continuous acoustic channel, instead of sending symbols directly (Figure 1). In each game round, a Speaker agent is tasked with conveying a single concept. The Speaker needs to explain this concept using a speech waveform which is transmitted over a noisy communication channel, and then received by a Listener agent. The Listener agent then classifies its understanding of the Speaker's concept. If the Speaker's target concept matches the classified concept from the Listener, the agents are rewarded. The Speaker is then presented with another concept and the cycle repeats.

Formally, in each episode, the environment generates $s$, a one-hot encoded vector representing one of $N$ target concepts from a set $\mathcal{S}$. The Speaker receives $s$ and generates a sequence of phones $c = (c_1, c_2, \ldots, c_M)$, each $c_t \in \mathcal{P}$ representing a phone from a predefined phonetic alphabet $\mathcal{P}$. The phone sequence is then converted into a waveform $w_{\text{raw}}$, an audio signal sampled at 16 kHz. For this we use a trained text-to-speech model (Black & Lenzo, 2000; Duddington, 2006). A channel noise function is then applied to the generated waveform, and the result $w_{\text{in}} = f(w_{\text{raw}})$ is presented as input to the Listener. The Listener converts the input waveform to a mel-scale spectrogram: a sequence of vectors over time representing the frequency content of an audio signal scaled to mimic human frequency perception (Davis & Mermelstein, 1980). Taking the mel-spectrogram sequence $X = (x_1, x_2, \ldots, x_T)$ of $T$ acoustic frames as input, the Listener agent outputs a vector $\hat{s}$ representing its predicted concept. The agents are both rewarded if the predicted word is equal to the target word $s = \hat{s}$.

To make the environment a bit more concrete, we present a brief example in Figure 2. For illustrative purposes, consider a set of concepts $\mathcal{S} = \{up, down, left, right\}$. The state representation for *down* would be $s = [0, 1, 0, 0]^\top$. A possible phone sequence generated by the Speaker would be $c = (d, a, ʊ, n, )$.[1] This would be synthesised, passed through the channel, and then be interpreted by the Listener agent. If the Listener's prediction is $\hat{s} = [0, 1, 0, 0]^\top$, then it selected the correct concept of *down*. The environment would then reward the agents accordingly:

$$r = \begin{cases} 1 & \text{if } s = \hat{s} \\ 0 & \text{otherwise} \end{cases} \tag{1}$$

In our environment we have modelled the task of the Speaker agent as a discrete problem. Despite this, the combination of both agents and their environment is a continuous communication task; in our communication channel, we apply continuous signal transforms which can be motivated by real acoustic environments. The Listener also needs to take in and process a noisy acoustic signal. It is true that the Speaker outputs a discrete sequence; what we have done here is to equip the Speaker with

---

[1]  and  respectively represent the start-of-sequence and end-of-sequence tokens.

articulatory capabilities so that these do not need to be learned by the model. There are studies that consider how articulation can be learned (Howard & Messum, 2014; Asada, 2016; Rasilo & Räsänen, 2017), but none of these do so in an RL environment, rather using a form of imitation learning. In Section 5 we discuss how future work could consider learning the articulation process itself within our environment, and the challenges involved in doing so.

# 3 LEARNING TO SPEAK AND HEAR USING RL

To train our agents, we use deep Q-learning (Mnih et al., 2013). For the Speaker agent, this means predicting the action-value of phone sequences. The Listener agent predicts the value of selecting each classification target $\hat{s} \in \mathcal{S}$.

## 3.1 SPEAKER MODEL

The Speaker agent is tasked with generating a sequence of phones $c$ describing a concept or idea. The model architecture is shown in Figure 3. The target concept is represented by the one-hot input state $s$. We use gated recurrent unit (GRU) based sequence generation as the core of the Speaker agent, which generates a sequence of Q-values, a distribution over phones $\mathcal{P}$ per output-step from 1 to $M$. The input state $s$ is embedded as the initial hidden state $h_0$ of the GRU. The output phone of each GRU layer is embedded as input to the next GRU layer.[2] We also make use of start-of-sequence (SOS) and end-of-sequence (EOS) tokens,  and  respectively, appended to the phone-set. These allow the Speaker to generate arbitrary length phone sequences up to a maximum length of $M$.

## 3.2 LISTENER MODEL

The Listener agent may be viewed as a classification task with the full model architecture illustrated in Figure 4. The model is roughly based on (Amodei et al., 2016). Given an input mel-spectrogram $X$, the Listener generates a set of state-action values. These action-values represent the expected reward for each classification vector $\hat{s}$.

We first apply a set of convolutional layers over the input mel-spectrogram, keeping the size of the time-axis consistent throughout. We then flatten the convolution outputs over the filters and feature axis, resulting in a single vector per time step. We process each vector through a bidirectional GRU, feeding the final hidden state through a linear layer to arrive at our final action-value predictions. An argmax of these action-values gives us a greedy prediction for $\hat{s}$.

---

[2]No gradients flow through the argmax: this connection indicates to the network which phone was selected at the previous GRU step.

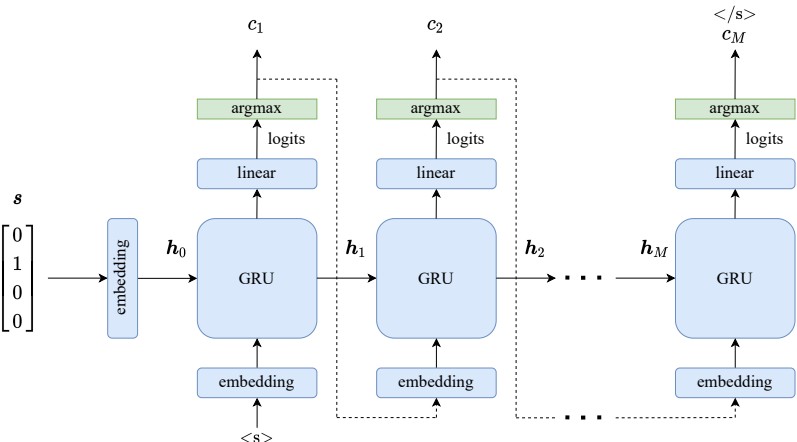

Figure 3: The Speaker agent generates an arbitrary length sequence of action-values given an input concept represented by $s$.

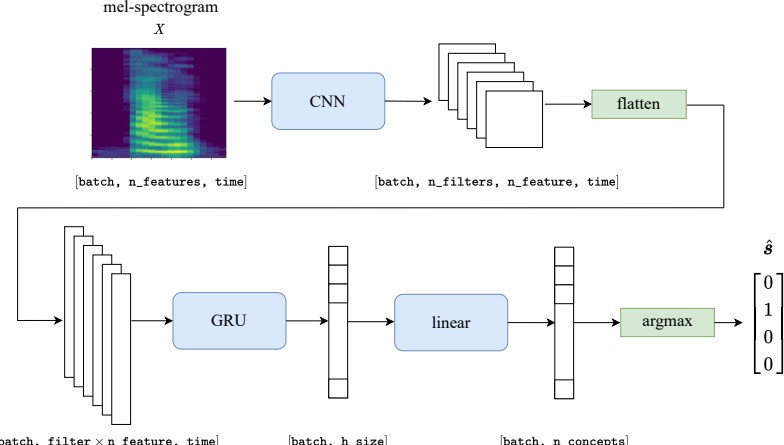

Figure 4: The Listener agent Q-network generates action-values given an input mel-spectrogram $X$.

### 3.3 DEEP Q-LEARNING

The Q-network of the Speaker agent generates a sequence of phones $c$ in every communication round until the EOS token is reached. The sequence of phones may be seen as predicting an action sequence per environment step, while standard RL generally only predicts a single action per step. To train such a Q-network, we therefore modify the general gradient-descent update equation from Sutton & Barto (1998). Since we only have a single communication round, we update the model parameters $\theta$ as follows:

$$\theta \leftarrow \theta + \alpha \left[ r - \frac{1}{M} \sum_{m=1}^{M} \hat{q}_m(S, A; \theta) \right] \nabla \hat{q}(S, A; \theta), \tag{2}$$

where the reward $r$ is given in (1), $S$ is the environment state, $A$ is the action, $\alpha$ is the learning rate, and $\hat{q} = (\hat{q}_1, \hat{q}_2, \dots, \hat{q}_M)$. For the Speaker, $\hat{q}_m$ is the value of performing the action $c_m$ at output $m$. For the Speaker, the environment state would be the desired concept $S = s$ and the actions would be $A = c = (c_1, c_2, ..., c_M)$, the output of the network in Figure 3.

The Listener is also trained using (2), but here this corresponds to the more standard case where the agent produces a single action, i.e. $M = 1$. Concretely, for the Listener this action is $A = \hat{s}$, the output of the network in Figure 4. The Listener's environment is the mel-spectrogram $S = X$. The Speaker and Listener each have their own independent learner and replay buffer (Mnih et al., 2013). A replay buffer is a storage buffer that keeps track of the observed environment states, actions and rewards. The replay buffer is then sampled when updating the agent's Q-networks through gradient descent with (2). We may see this two-agent environment as multi-agent deep Q-learning (Tampuu et al., 2017), and therefore have to take careful consideration of the non-stationary replay buffer: we limit the maximum replay buffer size to twice the batch size. This ensures that the agent learns only from its most recent experiences.

## 4 EXPERIMENTS

### 4.1 IMPLEMENTATION

The lossy communication channel has Gaussian white noise with a signal-to-noise ratio (SNR) of $30$ dB, unless otherwise stated. During training, the channel applies Gaussian-sampled time stretch and pitch shift using Librosa (McFee et al., 2021), with variance $0.4$ and $0.3$, respectively. The channel also masks up to $15\%$ of the mel-spectrogram time-axis during training. We train our agents with an $\epsilon$-greedy exploration, where $\epsilon$ is decayed exponentially from $0.1$ to $0$ over the training steps.

We use eSpeak (Duddington, 2006) as our speech synthesiser. eSpeak is a parametric text-to-speech software package that uses formant synthesis to generate audio from phone sequences. Festival (Black

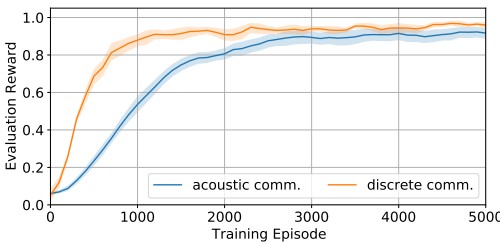 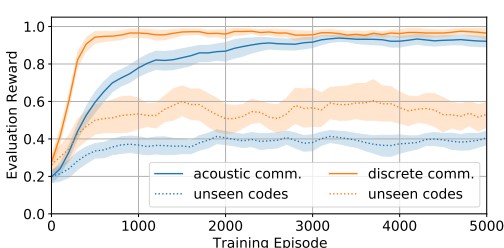

(a) Mean evaluation reward of the Listener agent interpreting a single concept over 20 runs.

(b) Mean evaluation reward of the Listener agent interpreting two concepts in each round.

Figure 5: Results for unconstrained communication. The agents are evaluated every 100 training episodes over 20 runs. Shading indicates the bootstrapped 95% confidence interval.

& Lenzo, 2000) was also tested, although eSpeak is favoured for its simpler phone scheme and multi-language support. We use eSpeak's full English phone-set of 164 unique phones and phonetic modifiers. The standard maximum number of phones the Speaker is allowed to generate in each communication round is $M = 5$, including the EOS token. All GRUs have 2 layers with a hidden layer size of 256. All Speaker agent embeddings (Section 3.1) are also 256-dimensional. The Listener (Section 3.2) uses 4 convolutional layers, each with 64 filters and a kernel width and height of 3. The input to the first convolutional layer is a sequence of 128-dimensional mel-spectrogram vectors extracted every 32 ms. We apply zero padding of size 1 at each layer to retain the input dimensions. Additional experimental details are given in Appendix A.

## 4.2 UNCONSTRAINED COMMUNICATION OF SINGLE CONCEPTS

**Motivation** We first verify that the environment works as expected and that a valid communication protocol emerges when no constraints are applied to the agents.

**Setup** The Speaker and Listener agents are trained simultaneously here, as described in Section 3.3. The agents are tasked with communicating 16 unique concepts. We compare our acoustic communication to a discrete baseline based on RIAL (Foerster et al., 2017). In this setup, the CNN of the Listener agent is replaced by an embedding network, allowing the discrete symbols of the Speaker to be directly interpreted by the Listener. The Speaker's discrete alphabet size of setup is equal to the phonetic alphabet size of 164. Improvements have been made to RIAL—e.g. (Eccles et al., 2019; Chaabouni et al., 2020)—although RAIL itself proves sufficient as a comparison to our proposed acoustic communication setting.

**Findings** Figure 5a shows the mean evaluation reward of the Listener agent over training steps. (This is also an indication of the Speaker's performance, since without successful coordination between the two agents, no reward is given to either.) The agents achieve a final mean reward of 0.917 after 5000 training episodes, successfully developing a valid communication protocol for roughly 15 out of the total of 16 concepts.[3] This is comparable to the performance of the purely discrete communication which reaches a mean evaluation reward of 0.959. What does the communication sound like? Since there are no constraints placed on communication, the agents can easily coordinate to use arbitrary phone sequences to communicate distinct concepts. The interested reader can listen to generated samples.[4] We next consider a more involved setting in order to study composition and generalisation.

## 4.3 UNCONSTRAINED COMMUNICATION GENERALISING TO MULTIPLE CONCEPTS

**Motivation** To study composition and generalisation, we perform an experiment based on (Kirby, 2001). They used an iterative language model (ILM) to convey two separate meanings ($a$ and $b$) in a single string. This ILM was able to generate structured compositional mappings from meaning to strings. For example, in one result they found $a_0 \rightarrow \texttt{q}$ and $b_0 \rightarrow \texttt{da}$. The combination of the two

---

[3]The maximum evaluation reward in all experiments is 1.0.

[4]Audio samples for all experiments are available at https://iclr2022-1504.github.io/samples/.

Table 1: Mean evaluation reward of the two-concept experiments with varying channel noise. The results for no lossy communication channel is also shown. The 95% confidence for all values falls within 0.01.

| Average SNR (dB) | Training Codes | Unseen Codes |
|---|---|---|
| no channel | **0.966** | 0.386 |
| 40 | 0.878 | 0.389 |
| 30 | 0.931 | 0.402 |
| 20 | 0.895 | **0.413** |
| 10 | 0.731 | 0.361 |
| 0 | 0.654 | 0.366 |

Table 2: Output sequences from a trained Speaker. Each entry corresponds to a combination of two concepts, $s_1$ and $s_2$, respectively. The bold combinations were unseen during training.

|  | | $s_1$ | | | |
|---|---|---|---|---|---|
| | | 0 | 1 | 2 | 3 |
| $s_2$ | 0 | nnʎɣɣx | ðʎʎççç | nsspxx | **nnsssss** |
| | 1 | jʎʎeee | əəʀʀeе | **wwwxxx** | sssəəə |
| | 2 | jjʎʎːː | **ðpʎʎj:** | ðwppçx | enɣsss |
| | 3 | **jjʎːːː** | ɣðð̃pːː | ɣjxxxp | ɣssːːː |

meanings was therefore $(a_0, b_0) \rightarrow$ qda. Similarly, $(a_1, b_0) \rightarrow$ bguda with $a_1 \rightarrow$ bgu. Motivated by this, we try to test the generalisation capabilities in continuous signalling in our environment.

**Setup** Rather than conveying a single concept in each episode, we now ask the agents to convey two concepts. The target concept $s$ and predicted concept $\hat{s}$ now become $s_1, s_2$ and $\hat{s}_1, \hat{s}_2$, respectively. We also make sure that some concept combinations are never seen during training. We then see if the agents are still able to convey these concept combinations at test time, indicating how well the agents generalise to novel inputs. The reward model is adjusted accordingly, with the agents receiving $0.5$ for each concept correctly identified by the Listener. Here $s_1$ can take on 4 distinct concepts while $s_2$ can take on another 4 concepts. Out of the 16 total combinations, we make sure that 4 are never seen during training. The unseen combinations are chosen such that there remains an even distribution of individual unseen concepts. We also increase the maximum phone length to $M = 7$. To encourage compositionality (Kottur et al., 2017), we limit the size of the phonetic alphabet to 16.

As an example, you can think of $s_1$ as indicating an item from the set of concepts $\mathcal{S}_1 = \{up, down, left, right\}$ while $s_2$ indicates and item from $\mathcal{S}_2 = \{fast, medium, regular, slow\}$ and we want the agents to communicate concept combinations such as *up+fast*. Some combinations such as *right+slow* is never given as the target concept combination during training (but e.g. *right+fast* and *left+slow* would be), and we see if the agents can generalise to these unseen combinations at test time and how they do it.

**Findings: Quantitative** The results are shown in Figure 5b. We see the mean evaluation reward of the acoustic Listener agent reaches $0.931$ on the training concept combinations. This is slightly lower than the discrete case which reaches a mean of $0.965$. The acoustic communication agents achieve a mean evaluation reward of $0.402$ on the unseen combinations, indicating that they are usually able to successfully communicate at least one of the two concepts. The discrete agents do marginally better on unseen combinations, with slightly higher variance. The chance-level baseline for this task would receive a mean reward of $0.25$. The performance on the unseen combinations is thus better than random.

Table 1 shows the mean evaluation reward of the same two-concept experiments, but now with varying degrees of channel noise expressed in SNR.[5] The goal here is to evaluate how the channel influences the generalisation of the agents to unseen input combinations. In the no-channel case, the Speaker output is directly input to the Listener agent, without any time stretching or pitch shifting. The no channel case does best on the training codes as expected, but does not generalise as well to unseen input combinations. We find that increasing channel noise decreases the performance of the training codes and increases generalisation performance on unseen codes, up to a point where both decrease. This is an early indication that the channel specifically influences generalisation.

Lazaridou et al. (2018) reported the structural similarity of the emergent communication in terms of Spearman $\rho$ correlation between the input and message space, known as topographic similarity or *topism* (Brighton & Kirby, 2006). Chaabouni et al. (2020) extended this metric by introducing two new metrics. Positional disentanglement (*posdis*) measures the positional contribution of symbols to

---

[5]The SNR is calculated based on the average energy in a signal generated by eSpeak.

Table 3: Compositionality metrics of the unconstrained multi-concept Speaker agents. The mean evaluation metrics and 95% confidence bounds are shown

|  | *topism* | *posdis* | *bosdis* |
|---|---|---|---|
| acoustic comm. | 0.265 ($\pm$0.041) | 0.103 ($\pm$0.015) | 0.116 ($\pm$0.018) |
| discrete comm. | 0.244 ($\pm$0.032) | 0.087 ($\pm$0.017) | 0.118 ($\pm$0.017) |

meaning. Bag-of-symbols disentanglement (*bosdis*) measures distinct symbol meaning but does so in a permutation-invariant language way. We record all 3 metrics for the case where the average SNR is 30 dB, taking measurements between the input space and the sequence of discrete phones. The results are shown in Table 3. For *topism*, we average 0.265, which is comparable to the results of (Lazaridou et al., 2018). For *posdis* and *bosdis*, we average 0.103 and 0.116, respectively. This falls within the lower end of the results of (Chaabouni et al., 2020). All three metrics yield similar results for both acoustic and discrete communication.

**Findings: Qualitative**   Table 2 shows examples of the sequences produced by a trained Speaker agent for each concept combination, with the phone units written using the international phonetic alphabet. Ideally, we would want each row and each column to affect the phonetic sequence in a unique way. This would indicate that the agents have learnt a compositional language protocol, combining phonetic segments together to create a sequence in which the Listener can distinguish the individual component concepts. We see this type of behaviour to some degree in our Speaker samples, such as the [x] phones for $s_1 = 2$ or the repeated [s] sound when $s_1 = 3$. This indicates at least some level of compositionality in the learned communication. More qualitatively, the realisation from eSpeak of [ʎ] sounds very similar to [n] for $s_2 = 0$. (We refer the reader to the sample page, linked in Section 4.2.)

The bold phone sequences in Table 2 were unseen during training. The agents correctly classified one combination ($s_1, s_2 = 3, 0$) out of the 4 unseen combinations. For the other 3 unseen combinations, the agents correctly predicted at least $s_1$ or $s_2$ correctly. These sequences also show some degree of compositionality, such as the [jʎ] sequence where $s_1 = 0$. We should note that the agents are never specifically encouraged to develop any sort of compositionality in this experiment. They could, for example, use a unique single phone for each of the 16 concept combinations.

## 4.4   GROUNDING EMERGENT COMMUNICATION

**Motivation**   Although the Speaker uses an English phone-set, up to this point there has been no reason for the agents to actually learn to use English words to convey the concepts. In this subsection, either the Speaker or Listener is predisposed to speak or hear English words, and the other agent needs to act accordingly. One scientific motivation for this setting is that it can be used to study how an infant learns language from a caregiver (Kuhl, 2005). To study this computationally, several studies have looked at cognitive models of early vocal development through infant-caregiver interaction; Asada (2016) provides a comprehensive review. Most of these studies, however, considered the problem of learning to vocalise (Howard & Messum, 2014; Moulin-Frier et al., 2015; Rasilo & Räsänen, 2017), which limits the types of interactions and environmental rewards that can be incorporated into the model. We instead simplify the vocalisation process by using an existing synthesiser, but this allows us to use modern MARL techniques to study continuous signalling.

We first give the Listener agent the infant role, and the Speaker will be the caregiver. This mimics the setting where an infant learns to identify words spoken by a caregiver. Later, we reverse the roles, having the Speaker agent assume the infant role. This represents an infant learning to speak their first words and their caregiver responds to recognised words. Since here one agent (the caregiver) has an explicit notion of the meaning of a word, this process can be described as "grounding" from the other agent's perspective (the infant).

**Setup**   We first consider a setting where we have a single set of 4 concepts $\mathcal{S} = \{up, down, left, right\}$. While this is similar to the examples given in preceding sections, here the agents will be required to use actual English words to convey these concepts. In the setting where the Listener acts as an infant, the caregiver Speaker agent speaks English words; the Speaker consists simply of a dictionary lookup for the pronunciation of the word, which is then generated by eSpeak.

In the setting where the Speaker takes on the role of the infant, the Listener is now a static entity that can recognise English words; we make use of a dynamic time warping (DTW) system that matches the incoming waveform to a set of reference words and selects the closest one as its output label. 50 reference words are generated by eSpeak. The action-space of the Speaker agent is very large ($|\mathcal{P}|^M$), and would be near impossible to explore entirely. Therefore, we provide guidance: with probability $\epsilon$ (Section 4.1), choose the correct ground truth phonetic sequence for $s$. We also consider the two-concept combination setting of Section 4.3 where either the Speaker or Listener now hears or speaks actual English words; DTW is too slow for the static Listener in this case, so here we first train the Listener in the infant role and then fix it as the caregiver when training the Speaker.

**Findings: Grounding the Listener**  Here the Listener is trained while the Speaker is a fixed caregiver. The Listener agent reached a mean evaluation reward of 1.0, indicating the agent learnt to correctly classify all 4 target words 100% of the time (full graphs given in Appendix B.1). The Listener agent was also tested with a vocabulary size of 50, consisting of the 50 most common English words including the original *up*, *down*, *left*, and *right*. With this setup, the Listener still reached a mean evaluation reward of 0.934.

**Findings: Grounding the Speaker**  We now ground the Speaker agent by swapping its role to that of the infant. The Speaker agent reaches a mean evaluation reward of 0.983 over 20 runs, indicating it is generally able to articulate all of the 4 target words. Table 4 gives samples of one of the experiment runs and compares them to the eSpeak ground truth phonetic descriptions. Although appearing very different to the ground truth, the audio generated by eSpeak of the phone sequences qualitatively similar. The reader can confirm this for themselves by listening to the generated samples (again we refer the reader to the sample page, linked in Section 4.2.)

**Findings: Grounding generalisation in communicating two concepts**  Analogous to Section 4.3, we now have infant and caregiver agents in a setting with two concepts, specifically $\mathcal{S}_1 = \{up, down, left, right\}$ and $\mathcal{S}_2 = \{fast, medium, regular, slow\}$. Here, these sets don't simply serve as an example as in Section 4.3, but the Speaker would now actually say "up" when it is the caregiver and the Listener will now actually be pretrained to recognise the word "up" when it is the caregiver. 4 combinations are unseen during training: *up-slow*, *down-regular*, *left-medium*, and *right-fast*. Again we consider both role combinations of infant and caregiver. Figure 6a shows the results when training a two-word Listener agent. The agent reaches a mean evaluation reward of 1.0 for the training codes and 0.952 for the unseen code combinations. This indicates that the Listener agent learns near-optimal generalisation. As mentioned above, for the case where the Speaker is the infant, the DTW-based fixed Listener was found to be impractical. Thus, we use a static Listener agent pre-trained to classify 50 concepts for each $s_1$ and $s_2$. This totals to 2500 unique input combinations. The results of the two-word Speaker agent are shown in Figure 6b. The Speaker agent does not perform as well as the Listener agent, reaching a mean evaluation reward of 0.719 for the training word combinations and 0.425 for the unseen.

We have replicated the experiments in this subsection using the Afrikaans version of eSpeak, reaching similar performance to English. This shows our results are not language specific.

## 5  DISCUSSION

The work we have presented here has gone further than Gao et al. (2020), which only allowed segmented template words to be generated: our Speaker agent has the ability to generate unique audio waveforms. On the other hand, our Speaker can only generate sequences based on a fixed

Table 4: Table of the target word, ground truth phonetic description, and trained Speaker agent's predicted phonetic description.

| Target word | Ground truth | Predicted phones |
|:---:|:---:|:---:|
| *up* | ʌp | ʌvb |
| *down* | daʊn | daʊ |
| *left* | lɛft | lɛ |
| *right* | ɹaɪt | ɹaɪʃjn |

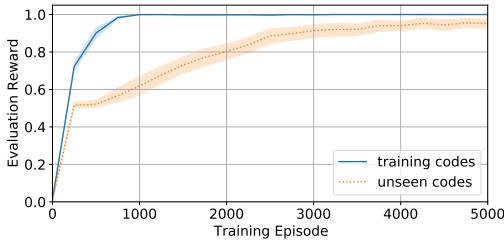 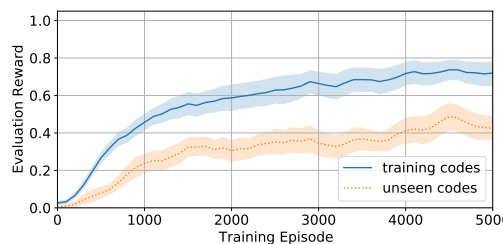

(a) Mean evaluation reward of two-word Listener agent over 20 training runs.

(b) Mean evaluation reward of two-word Speaker agent over 20 training runs.

Figure 6: Evaluation results of the grounded two-word Speaker and Listener agent during training. The mean evaluation reward of the unseen word combinations are also shown.

phone-set (which is then passed over a continuous acoustic channel). This is in contrast to earlier work (Howard & Messum, 2014; Asada, 2016; Rasilo & Räsänen, 2017) that considered a Speaker that learns a full articulation model in an effort to come as close as possible in imitating an utterance from a caregiver; this allows a Speaker to generate arbitrary learnt units. We have thus gone further than Gao et al. (2020) but not as far as these older studies. Nevertheless, our approach has the benefit that it is formulated in a modern MARL setting: it can therefore be easily extended. Future work can therefore consider whether articulation can be learnt as part of our model – possibly using imitation learning to guide the agent's exploration of the very large action-space of articulatory movements.

In the experiments carried out in this study, we only considered a single communication round. We also referred to our setup as multi-agent, which is accurate but could be extended even further where a single agent has both a speaking and listening module, and these composed agents then communicate with one another. Future work could therefore consider multi-round communication games between 2 or more agents. Such games would extend our work to the full MARL problem, where agents would need to "speak" to and "hear" each other to solve a common task.

Finally, in terms of future work, we saw in Section 4.3 the importance of the channel for generalisation. Adding white noise is, however, not a good enough simulation of real-life channel acoustic channels. But our approach could be extended with real background noise and more accurate models of environmental dynamics. This could form the basis for a computational investigation of the effect of real acoustic channels in language learning and emergence.

We reflect on our initial research question: Are we able to observe emergent language between agents with a continuous acoustic communication channel trained through RL? This work has laid only a first foundation for answering this larger question. We have showcased the capability of a environment and training approach which will serve as a means of further exploration in answering the question.

## ETHICS STATEMENT

We currently do not identify any obvious reasons to have ethical concerns about this work. Ethical considerations will be made taken into account in the future if some of the models are compared to data from human studies or trials.

## REPRODUCIBILITY STATEMENT

We provide all model and experimental details in Section 4.1, and additional details in Appendix A. The information given should provide enough details to reproduce these results. Finally, our code will be released on GitHub with an open-source license upon acceptance.

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

## APPENDICES

## A  EXPERIMENT DETAILS

### A.1  GENERAL EXPERIMENTAL SETUP

Here we provide the general setup for all experimentation.

| Parameter | Value |
|---|---|
| *Optimiser* | Adam |
| *Batch Size* | 128 |
| *Replay size* | 256 |
| *Training Episodes* | 5000 |
| *Evaluation interval* | 100 |
| *Evaluation episodes* | 25 |
| *Runs (varying seed)* | 20 |
| *GPU* | Nvidia RTX 2080 Super |
| *Time (per run)* | $\approx 30$ minutes |

### A.2  EXPERIMENT PARAMETERS

Here we provide specific details on a per-experiment basis. The phone sequence length $M$ in the grounded experiments is chosen such that the full ground truth phonetic pronunciation could be made by the speaker agent.

| Experiment | Agent | Learning Rate | Phone length ($M$) | GRU hidden size |
|---|---|---|---|---|
| Unconstrained Single-Concept | Speaker | $1 \times 10^{-4}$ | 5 | 256 |
| | Listener | $5 \times 10^{-5}$ | - | 256 |
| Unconstrained Multi-Concept | Speaker | $1 \times 10^{-5}$ | 7 | 512 |
| | Listener | $5 \times 10^{-5}$ | - | 512 |
| Grounded Single-Concept | Speaker | $1 \times 10^{-4}$ | 6 | 256 |
| | Listener | $5 \times 10^{-5}$ | - | 256 |
| Grounded Multi-Concept | Speaker | $1 \times 10^{-5}$ | 16 | 512 |
| | Listener | $5 \times 10^{-5}$ | - | 512 |

## B  RESULTS

### B.1  GROUNDING EMERGENT COMMUNICATION

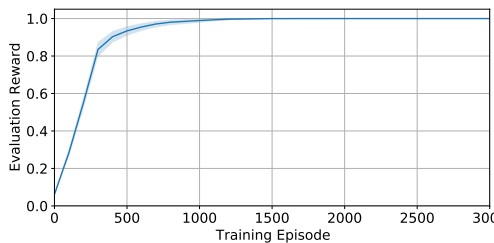

(a) Mean evaluation reward of Listener agent over 20 training runs.

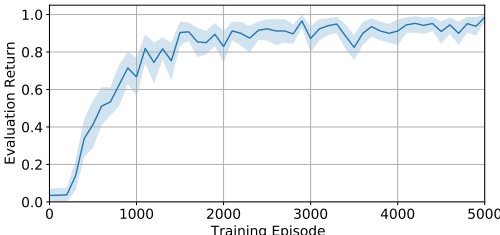

(b) Mean evaluation reward of Speaker agent over 20 training runs.

Figure 7: Evaluation results of the grounded Speaker and Listener agent during training. Shading indicates the bootstrapped 95% confidence interval.

