# OpenReview forum: "Towards Learning to Speak and Hear Through Multi-Agent Communication over a Continuous Acoustic Channel"
_ICLR.cc/2022/Conference — ICLR 2022 Submitted_

### Official Review · Reviewer_42Xh · 2021-10-16

**Correctness:** 4
**Technical Novelty And Significance:** 2
**Empirical Novelty And Significance:** 2
**Recommendation:** 5
**Confidence:** 4

**Main Review:**

Good points:
- pushes emergent communication in a new direction: that of oral communications

Bad points:
- some key missed references, i.e. Foerster et al 2016 section 6.4 uses a continuous channel, and finds that adding noise to the channel is necessary in order to effectively 'discretize' the outputs. I also feel that Lazaridou et al 2018 is a better reference than Chaabouni et al 2020, since Lazaridou et al 2018 was more or less the first work to use a referential task in the recent wave of emergent communication literature.
- a lot of the space in the paper appears to me to be used to describe relatively vanilla concepts, such as the architecture of the sender and receiver RNN models, what a replay buffer is, and some of the earlier experiments are I feel more like 'debugging'/smoke-test experiments, that show the architecture learns
- I wasn't sure what I was supposed to learn from the grounded experiments. I felt that it was unsurprising that grounding one agent in English would cause the other agent to learn a language that learns English
- the text to speech synthesize is pretrained, which goes I feel against the spirit of emergent communication. I understand that motivation for this was provided, and that motivation makes sense to me, but listing this for completion
- the meaning/object spaces being used are very small (1 dimension, of size 16; or two dimensions, each of size 4; for example)
- code is not provided with the submission
- I didn't come away with any sense of 'oh that was interesting/surprising; I learned something'

## Notes

Notes I made as I read through the paper:

### Abstract

I like the idea of using an accoustic channel :) It's something I've been intending to write a paper on sometime, just didn't get around to it yet :)

It seems clear to me that spoken language preceded written language, so starting with accoustic language, for emergent communication, seems like a great idea to me :)

I'm a little surprised/unclear why we map from symbols to accoustic. why not just keep everything continuous?

noisy communication channel being needed for compositionality and/or generalization has been shown in a few previous works.

### 1. Introduction

Mising reference "Learning to Communicate with Deep Multi-Agent Reinforcement Learning", Foerster et al 2016, which in section 6.4, uses a continuous channel, and discovers that by adding noise, the channel effectively becomes discretized.

Citing Chaabouni et al 2020 for 'referential signaling game' seems I feel not the original reference. "Emergence of linguistic communication from referential games with symbolic and pixel input", Lazaridou et al, 2018 was one of the first works to bring the idea of a referential game into emergent communication domain. Prior to that, the Lewis signaling game, Lewis, 1969, is, as far as I know, the original reference.

Why use deep Q-learning? Not that it's not a good choice, but curious why you choose this, when the vast majority of emergent communication works use simple REINFORCE (or Gumbel).

### 2. Environment

Using a pre-trained text-to-speech synthesizer seems to go against the goals of 'emergent' communication?

The explanation for why this is (later in the section) does make sense to me though. Baby steps :)

### 3. Learning to speak and hear using RL

A large amount of space (~1.5 pages) is being spent describing what appear to be quite vanilla sender and receiver models. I'm not suggesting that the models should be made 'fancier', but that that space could perhaps be filled with additional experiments, or analysis, etc.

### 4. Experiments

I feel discussing eSpeak vs Festival gives a sort of 'engineer-y' feel to the paper; and could be omitted.

#### 4.2 Unconstrainted communication of single concepts

I realize at this point that somehow I missed what are the actual input data. Ok, skimming right back to page 3, it looks like there is only a single meaning dimension, of size N? This could represent eg 'color'. From 4.2, it looks like N is 16, though it's not stated clearly. This is I feel an extremely small meaning/concept/object space.

On the whole, 4.2 seems like a superfluous debugging section, which doesn't really provide any information/signal, and could be omitted.

#### 4.3 Unconstrained communication generalizing to multiple concepts

I was briefly excited by the mention of Kirby's ILM, but I feel that the citation of Kirby's ILM is superfluous here: the experiment is simply changing from a 1-dimensional meaning space of size 16, to a 2-dimensional meaning size, where each dimensino is of size 4.

Again, this experiment seems to me to be quite vanilla.

I feel that in addition to providing a table of utterances to eye-ball, it could be slightly more rigorous to use objective measures of compositionality such as topological similarity.

I feel that the effect of channel noise in Table 2 appears quite weak to me. In addition, it could be argued that the noise is simply providing training time regularization, which we see as a drop in performance of the training codes, rather than leading to a qualitative shift in the emergent communication, as Foerster et al 2016 showed, in their section 6.4.

#### 4.4 Grounding emergent communication

I feel that 'grounded emergent communication' is a contradiction in terms: either the language is grounded, or it emerges. If there is any grounding provided, then it is a grounded task, not an emergent task, I feel?

I'm not really sure of the purpose of this set of experiments. The various agents are being grounded in english in various ways, and end up outputing some English-like language. I'm not really sure what I can learn from this?

I feel that using English-language grounding in a paper positioning itself as emergent communication is somewhat risky. With sufficient motivation and analysis, I don't have any fundamental objection to doing this though. Preference to come away with some sense of 'that was interesting; I learned something'. This might mean making the experiments challenging enough so that they do *not* actually work, and then digging into why, and what we can learn from that, perhaps?

**Summary Of The Paper:**

The paper proposes to move in the direction of emerging oral communication, instead of written communication. They create an architecture where the sender agent first outputs a string of symbols, which map statically to a fixed table of phonems. The sender uses an off-the-shelf pre-trained text-to-speech synthesizer to convert these phonemes into sounds. The sounds pass through a channel with Gaussian white noise added, along with Gaussian-sampled time-shift and pitch-shift. The receiver converts the input sounds into a mel-spectrogram, which is a sequence of vectors over time representing the frequency distribution. The receiver passes these vectors into a CNN, an RNN, and finally a linear classifier.

The paper motivates using the pre-trained speech-to-text synthesizer as a necessary initial baby-step, to keep the problem tractable in the context of RL.

The paper uses an encoder-decoder task, where the input is a symbolic concept comprising one dimension of size 16, or two dimensions of size 4. The output is a prediction of the original input concept.

The paper shows in a first experiment that their model can learn to predict the input concept correctly with a meaning space of 1 dimension of size 16. Then, in a second experiment, the concepts are extended to two dimensions, each of size 4. In addition, in this second experiment, the effect of communication channel noise is measured: adding noise to the channel decreases performance on concepts seen during training, but improves performance on a holdout set of novel input concepts.

In the final experiment, where the speaker or the listener are fixed, and grounded in English. In this case, the other agent will learn a language that is close to English.

**Summary Of The Review:**

On the whole, I like that the work pushes towards moving emergent communication into a new direction of oral communication. I feel that this work as it stands is an excellent start in this direction. However, I feel that the experiments and analysis could be extended somewhat, into scenarios where it feels somewhat less predictable what the outcomes will be. For example, what happens if the meaning space is larger? Does that work? If it doesn't work, how does it fail and why?

---

> ### Author Response · Authors · 2021-11-17
> **Response to Reviewer 42Xh**
>
> We really appreciate the detailed review and constructive feedback. We also appreciate the interest on the general research direction of acoustic communication.
>
> > i.e. Foerster et al 2016 section 6.4 uses a continuous channel, and finds that adding noise to the channel is necessary in order to effectively 'discretize' the outputs
>
> We would like to differentiate between the continuous channel that we’re using and that of prior work. In prior work, such as Foerster et al 2016, the communication channel which is a continuous fixed-length vector serves as a sort of common “hidden state” between agents trained in a centralised approach. Gradients are allowed to flow between agents through this common communication channel during training. This is quite different to our approach, where we have decentralised training and an acoustic channel with audio-based communication. Our channel has variable length and is presented as a sampled audio signal. With our setup, gradients cannot flow through this audio channel.
>
> > the text to speech synthesize is pretrained, which goes I feel against the spirit of emergent communication. I understand that motivation for this was provided, and that motivation makes sense to me, but listing this for completion
>
> What we may consider as future work is allowing the model to learn articulation itself. We could do this by modelling a human vocal tract and allowing an agent to control the articulation of sounds.
>
> > it could be slightly more rigorous to use objective measures of compositionality such as topological similarity.
>
> We have included quantitative metrics of topism (Brighton and Kirby, 2006), along with posdis and bosdis (Chaabouni et al 2020). We find our results comparable to that of Lazaridou et al 2018.

---

> > ### Comment · Reviewer_42Xh · 2021-11-23
> > **Thank you for the response**
> >
> > Thank you for the response. I feel that you have addressed one or two of my minor concerns, but have not addressed my over-arching concern that the work simply does not go into enough depth, I feel. Many of the experiments do not have surprising outcomes, I feel. The network learns. I don't really come away with a sense of 'oh, that was surprising, interesting, I learned something'.

---

### Official Review · Reviewer_6auy · 2021-10-28

**Correctness:** 2
**Technical Novelty And Significance:** 1
**Empirical Novelty And Significance:** 1
**Recommendation:** 3
**Confidence:** 4

**Main Review:**

This paper describes some experiments to simulate language acquisition in which a speaker talks to a listener over a continuous voice channel.  The speaker uses a GRU conditioned by one or more concepts to generate a sequence of sounds selected from an inventory of 160 English phonemes released by eSpeak.  The listener applies a convolutional net to the spectrogram of received sounds, then decodes sequential frames to generate a distribution over phonemes from which an argmax generates the decoded concept.   Learning uses a simple 1-step DQN.  The channel between speaker and listener has additive noise and time/pitch shifts applied.

The experiments include the unconstrained transmission of 1 and 2 concepts and the same but constrained so that either the speaker knows English words or the listener knows how to recognise English words.  Claims made include the ability to use composition and generalisation when moving from 1 to 2 concept transmission.

There are many things that I do not understand about this paper.

Firstly, when the speaker is generating a sequence of M phonemes, each phone is regarded as an action but the optimisation treats this an M-armed bandit problem and simply averages the Q values over all actions.  Why is this not treated as an M-step trajectory and solved using the standard DQN optimisation?

Secondly, it appears that all phoneme sequences are exactly M long, rather than a maximum of M long (see eg Table 1).  Is this true?   Wouldn't it be better to allow variable length and perhaps use the reward to encourage brevity.

Thirdly, how are the input concepts coded?  The examples suggest that each concept is coded as a 1-hot vector.  For multiple concepts, the natural assumption would be that the concepts are simply concatenated.  If this is the case, how does the listener work since it is only capable of forming a distribution over 1 set of concepts.  How does the listener distinguish multiple component concepts?

The claim that the results in Table 1 demonstrate the ability to learn a compositional language protocol are unconvincing.  Quoting 0.25 as the chance mean reward ignores the fact that the agents could simply recognise one of the concepts whenever the combination was unseen.

Adding noise will often improve the robustness of a classifier, why is the generalisation implied by Table 2 any different?  What is the statistical significance of the numbers in this table?

On page 9, it is claimed that using DTW for the 2 word listener case is impractical.  Why is this? If the Listener only has to consider a vocabulary of 50 words, there are only 2500 two word combinations which should be tractable for a DTW style recogniser with pruning.


**Summary Of The Paper:**

This paper describes some experiments to simulate language acquisition in which a speaker talks to a listener over a continuous voice channel.  The speaker learns to generate a phone sequence for each input concept that it wants to convey and the listener learns to recognise the input phone sequences to decode the embedded concept(s).  The speaker and listener use RL to learn. When the listener correctly identifies a concept both the speaker and listener receive a positive reward.  The authors claim that the system demonstrates the ability to exploit composition and generalisation.


**Summary Of The Review:**

I am not convinced by the motivation of this paper, the presentation is poor and the experimental results are unconvincing.  The experimental design appears to involve some unnecessary approximations.  The paper claims to be an advance on Gao, but Gao addresses the problem of segmenting a speech stream into words when the listener does not have a pronouncing dictionary.  Here the use of the continuous channel appears to me to be an unnecessarily complicated way of adding channel noise to what is otherwise a discrete system.

---

> ### Author Response · Authors · 2021-11-17
> **Response to Reviewer 6auy**
>
> We really appreciate the response and constructive feedback. We have responded to the primary concerns below.
>
> > Firstly, when the speaker is generating a sequence of M phonemes, each phone is regarded as an action but the optimisation treats this an M-armed bandit problem and simply averages the Q values over all actions. Why is this not treated as an M-step trajectory and solved using the standard DQN optimisation?
>
> The primary reason is the sequence of phones needs to be synthesized all at once. With most synthesis methods, you can't partially synthesize a waveform. Secondly, we need this setup if we want multi-round communication in future work. This means that in each time-step in the environment, the agent will have to output a whole phone sequence in one inference step before moving to the next time-step.
>
> > Secondly, it appears that all phoneme sequences are exactly M long, rather than a maximum of M long (see eg Table 1). Is this true? Wouldn't it be better to allow variable length and perhaps use the reward to encourage brevity.
>
> The agents are actually allowed to output variable length sequences as explained in Section 3.1: “...allow the Speaker to generate arbitrary length phone sequences up to a maximum length of $M$”. Although, the agents preferred to use the entire sequence length in most two-concept experiments. This happened to be the case in Table 1, where all phoneme sequences were exactly M long. We appreciate the idea of including a reward penalty for longer sequences and to see if the agent learns brevity, and will be considered in future work.
>
> > Thirdly, how are the input concepts coded? The examples suggest that each concept is coded as a 1-hot vector. For multiple concepts, the natural assumption would be that the concepts are simply concatenated. If this is the case, how does the listener work since it is only capable of forming a distribution over 1 set of concepts. How does the listener distinguish multiple component concepts?
>
> It is correct that the concepts are one-hot-encoded, as stated in Section 3.1: “...target concept is represented by the one-hot input state $\boldsymbol s$”. It is also correct that input concepts are concatenated in the case of multiple concepts. The listener is actually adjusted to have two separate action heads. One head to determine the first concept, and another to determine the second concept. Alternatively, this may be replaced with an RNN to generate an arbitrary number of concepts.
>
> > On page 9, it is claimed that using DTW for the 2 word listener case is impractical. Why is this? If the Listener only has to consider a vocabulary of 50 words, there are only 2500 two word combinations which should be tractable for a DTW style recogniser with pruning.
>
> While we agree that this is tractable for the case where we consider 2500 combinations, this was a compute constraint on our end. In the single-concept experiments, it took approximately 5 hours to run 20 runs of 5000 episodes each, primarily limited by the DTW computation. Scaling this to the two-concept case, it would take approximately 250 hours. While this is doable, we were limited by compute constraints. We are also concerned about DTW in terms of scalability and practicality for future work, where we may not necessarily have set reference combinations. If we consider three or four-concept communication with a vocabulary size of 50, we’d already be looking at 125,000 to 6,250,000 reference combinations.
>
> > Here the use of the continuous channel appears to me to be an unnecessarily complicated way of adding channel noise to what is otherwise a discrete system.
>
> While this is valid criticism, we would like to move in a direction of study focused on the continuous acoustic channel's emergent characteristics, bridging multi-agent communication and speech processing. Without this channel, there is nothing to study.

---

> > ### Comment · Reviewer_6auy · 2021-11-22
> > **Review unchanged**
> >
> > I appreciate the authors providing clear answers to my questions.  However, I remain unconvinced by some of the answers and the experimental setup generally.  I see this as essentially a discrete channel experiment with a number of compromises, and I didn't get a clear takeaway message from the paper.

---

### Official Review · Reviewer_yr56 · 2021-11-02

**Correctness:** 4
**Technical Novelty And Significance:** 3
**Empirical Novelty And Significance:** 1
**Recommendation:** 3
**Confidence:** 4

**Main Review:**

## Strengths
- The environment is novel and provides a good basis for studying certain traits of continuous-channel referential games.
- Excellent clarity and presentation of ideas

## Weaknesses
- The experiments provided do not analyze the unique characteristics of the environment introduced; instead, the experiments are similar to the typical gamut for a discrete symbol-based referential game.
- The analysis of compositionality given is insufficient.
    It appears that a single example was analyzed qualitatively.
    In the presence of definitive results, this might be acceptable, but in its absence, as in this paper, there needs to be quantitative analysis across multiple runs to demonstrate the robustness of the phenomenon.


## Recommendation
I recommend rejecting the paper.
While I appreciate the environment introduced and believe it to be promising, the experiments presented fail to highlight the novelty of the environment.
It is not the case that I find the experimental results inadequate, rather, the experiments run in the first place are generic and do not illustrate the points of interest with a continuous-channel referential game.
It is possible that I could be convinced otherwise as I truly appreciate the environment, but I would need to see an argument as to how the paper properly supports its introduction of a new environment.


## Justification
As my primary critique concerns the experiments, I will address each individually mentioning any deficiencies and potential improvements.
- unconstrained, single-concept:
    This is adquate to demonstrate that a minimal environment works (as stated in the paper).
- unconstrained, multi-concept:
    This needs a direct comparison to a traditional discrete-channel referential game.
    Thus, the experiment could answer questions such as:
    - Does compositionality emerge at the same rate in continuous- and discrete-channel games?
    - What is it about an audio channel model that makes compositionality easier or more difficult to learn?
    - Does the interaction between various phonemes restrict the means by which compositional codes can be communicated (e.g., /aio/ and /aeo/ are harder to distinguish than /abo/ and /ato/)?
- multi-concept and noise:
    The noise you add to the channel has a number of different components; there should be an ablation study to illustrate the effects of these components.
    If the experiment does not peer inside the structure of the noise and treats it as a gray box instead, it is not much different than a discrete-channel environment where random edits are made to the message.
- grounded language learning:
    In both of these experiments, there is analysis provided on what aspects of the audio-based communication channel make the problem harder, easier, or just different from the same experiment with a discrete channel.
    For example, experiments could investigate if using words which are phonologically similar (e.g., "boat" and "moat") is harder to distinguish than dissimilar words.
    Or an experiment could address if using the phonotactics from a natural language results is more effective than using a randomly selected phonemes due to necessity for natural languages to have acoustically distinct words.


Specific criteria:
- Correctness: 4
    - The claims are supported, but I do not think the claims go far enough (e.g., "noise has an effect on generalization" is claimed when instead what that effect is needs to be characterized).
- Technical Novelty and Significance: 3
- Empirical Novelty and Significance: 1

## Questions
_What_ is unique about the environment has been presented well, so my primary question is _how_ is it unique?
In other words, I see that the structure of the environment is unique, but what are the consequences which this structural difference begets?

## Additional Comments
- `s1`: Introduction is paced well
- `s2p2`: what is T? Is it just every tick of the audio signal?
- `s3.2p2`: What is the specific reasoning for using a bidirectional GRU? Using a model capable of streaming would make sense from the point of view of inductive biases.
- `s3.2 p2`: Due to the argmax, is it the case that the "confidence" of the predictions is not used in optimization? This seems like it would artificially make optimization harder.
- `Table 2`: Are these values computed over a single or multiple runs (in which case include stddev/confidence intervals).
- `s4.3 Findings p4`: "the channel ... influences generalisation" -- Without further explication, this potentially interesting point does not gain any traction.


**Summary Of The Paper:**

This paper tackles the well-established signalling emergent language game with the innovation of using a continuous, audio-based channel for communication as opposed to the more typical discrete, symbolic channel.
This continuous channel takes the form of a sender generating phones which are then synthesized into an audio waveform; this audio is then passed to the receiver after being transformed into a spectrogram.
The experiments then evaluate some general characteristics of the aforementioned referential game.

The following empirical evaluations are performed:
- Basic referential game
- 2-attribute ref. game to test for generalization and compositionality
- Varying the noisiness of the channel to determine effects on generalization
- Grounding the sender in English and letting the receiver learn
- Grounding the receiver in English and letting the sender learn

The primary contributions of the paper are:
- Presenting a phone-based, continuous-channel referential game
- The effect of channel noise on generalization
- Ability to learn a grounded language from a pretrained conversation partner


**Summary Of The Review:**

The paper presents a well-designed, novel referential game where the communication channel is continuous and audio-based.
The paper fails to demonstrate with its experiments the actual effects of using this continuous channel.
As a result, my decision is "reject"; I think a new set of experiments highlighting the novel setup will yield a strong paper.

---

> ### Author Response · Authors · 2021-11-17
> **Response to Reviewer yr56**
>
> We really appreciate the interest in our environment and the general research direction. Your remarks related to the deficiencies in our experimentation are very helpful, and will help guide our current and future research.
>
>
> > The analysis of compositionality given is insufficient.
>
> We have included a paragraph in the experimental section 4.3. Here we included quantitative metrics of topism (Brighton and Kirby, 2006), along with posdis and bosdis (Chaabouni et al 2020). We find our results comparable to that of Lazaridou et al 2018.
>
> > This needs a direct comparison to a traditional discrete-channel referential game
>
> We have updated the paper to include a comparison to a discrete baseline. We also want to restate, especially given the inclusion of a discrete communication baseline, that our goal is not to achieve state-of-the-art performance in discrete communication, as e.g. in the work of (Eccles et al. 2019; Chaabouni et al. 2020). Instead our focus is in developing a new direction towards acoustic communication more akin to that of humans.
>
> > What is the specific reasoning for using a bidirectional GRU? Using a model capable of streaming would make sense from the point of view of inductive biases.
>
> In our specific case, causal operations over the communication channel are not so important. We actually found no difference in results when a bidirectional versus a unidirectional GRU. Although in future, it may make more sense to use a unidirectional GRU.
>
> > Due to the argmax, is it the case that the "confidence" of the predictions is not used in optimization? This seems like it would artificially make optimization harder.
>
> This is possible, it may help stabilise during inference, although we didn’t actually test for this. During inference it allows the agent to know exactly which phone was selected in the previous step.

---

> > ### Comment · Reviewer_yr56 · 2021-11-22
> > **Review unchanged**
> >
> > Thank you for the rebuttal.
> >
> > I think the comparisons that were added to a "typical" discrete channel model are the right direction, but I will reiterate the point in my review that I do not believe the analyses performed highlights the unique characteristics of an acoustic channel.
> > I am entirely in accord with the rebuttal's saying that "out goal is to achieve state-of-the-art performance in discrete communication... Instead our focus is in developing a new direction towards acoustic communication more akin to that of humans."  (for one, "SotA" is practically meaningless in emergent language).
> > The issue I see is that this "focus in developing a new direction" is achieved with the environment introduced but _not_ with the empirical analysis due to the aforementioned reasons.

---

### Official Review · Reviewer_GZXE · 2021-11-02

**Correctness:** 3
**Technical Novelty And Significance:** 2
**Empirical Novelty And Significance:** 2
**Recommendation:** 3
**Confidence:** 5

**Main Review:**

The paper explores a two-agent communication problem where the Sender agent sends a continuous waveform to the Listener who is tasked to reconstruct the original concept vector. There are two settings of concepts used, single concepts (or a flat vector) and double concepts (consists of a combination of two separate meanings).

However, there are many important details missing in the main text. It is not clear how big is the alphabet size that is used by the Speaker to construct the waveform. It is an important parameter as it affects the compositional generalization of the agents (Chaabouni et al. 2020, Kottur et al. 2017, Resnick et al. 2020, Li and Bowling 2019).
Regardless of the alphabet size used, the problem setting used is quite 'toyish'. Even in the double concepts game the total possible combination of all concepts is 16. In most of the previous literature in this field, even the minimal setup uses 3 concepts of 5 types each (Chaabouni et la. 2020) and then results are validated using a more realistic problem setting either using real images as input or increasing the input space. The question here is would this method scale to larger problem settings.

The idea of using signalling to make one agent learn about the underlying concept and active listening to help the listener solve the given downstream task is already explored in Eccles et al 2019. Unfortunately, there has been no comparisons made with this work that seems to be the most relevant to the proposed method.

Moreover, many relevant are not cited in this work (for example the above works). I would encourage the authors to do a thorough literature survey of the recent work done in the space of compositionality and generalization in emergent languages.

The hypothesis around compositionality is only evaluated qualitatively. There have been some good progress done to compute domain independent quantitative compositionality metrics that are well established and grounded (Chaabouni et al. 2020, Lazaridou et al. 2018, Resnick et al. 2020, Andreas 2019). In addition, the notion of compositionality captured in the paper is more aligned to combinatorial generalization.

Good-enough compositional data augmentation. Andreas 2019
Entropy minimization in emergent languages. Kharitonov et al. 2020
Compositionality and generalization in emergent languages. Chaabouni et al. 2020
Ease-of teaching and language structure from emergent communication. Li and Bowling 2019
Capacity, bandwidth, and compositionality in emergent language learning. Resnick et al. 2020
Natural language does not emerge ‘naturally’ in multi-agent dialog. Kottur et al. 2017
Emergence of linguistic communication from referential games with symbolic and pixel input. Lazaridou et al. 2018

**Summary Of The Paper:**

The paper proposes a method to learn meaningful emergent languages with a continuous communication channel. The game consists of two agents that perform a unidirectional communication where the agents are trained using deep RL. The training follows an iterated procedure where one agent acts as a 'teacher', who has the actual meaning of the word, and the other as a 'student', who is trained at a given iteration by using information obtained form the teacher during training. The authors show that this type of grounding allows generalization and helps in learning a compositional language.

**Summary Of The Review:**

The paper explores an interesting research question to train agents that can communicate using a continuous communication channel and be able to solve the given downstream task thereby exhibiting compositional generalization. Although the underlying hypothesis is interesting, the idea is not completely novel and not evaluated extensively using well-established metrics in the literature. This work is a good preliminary work in investigating this research direction and I would encourage the authors to refine the evaluation and make explicit comparisons with prior work in the future submissions of this work.

---

> ### Author Response · Authors · 2021-11-17
> **Response to Reviewer GZXE**
>
> We really appreciate the detailed review and constructive feedback. We especially appreciate the pointers to related literature that we missed, and also the positive remarks on the general research direction.
>
> >  It is not clear how big is the alphabet size that is used by the Speaker to construct the waveform
>
> Do you mean the size of the phonetic alphabet? If so, we do specify that “we use eSpeak’s full English phone-set of 164 unique phones” in Section 4.1, although we may not have been clear enough. If there is still anything unclear or we have misunderstood your question, please let us know.
>
> > The question here is would this method scale to larger problem settings.
>
> We agree that experiments towards scaling our environment and problem setting would be valuable and indeed that is the focus of our current work.  But given the challenges in combining concepts from MARL and speech processing, we believe that this work can stand on its own as a first step towards the grander goal of seeing how large-scale continuous communication can emerge between multiple agents.
>
> > already explored in Eccles et al 2019
>
> We are not sure which paper is referenced here. Is it https://arxiv.org/pdf/1912.05676.pdf? If so, we do not believe this paper actually performs communication over an acoustic channel. The paper focuses on improving decentralised discrete communication by introducing positive biases. While it may provide a useful baseline comparison as a discrete system, our work is fundamentally different with a focus on communication through an acoustic channel.
>
> We hope this helps to clear up some misunderstanding, if not please let us know. We are very appreciative of the feedback.

---

> > ### Comment · Reviewer_GZXE · 2021-11-22
> > **Response**
> >
> > I thank the authors for answering some of my questions. Indeed, the paper I was referring to is the one mentioned by the authors. I apologize for missing the citation.
> >
> > The authors argue that Eccles et al. did not use an acoustic channel for communication. I agree that is the case but my concern wasn't related to the mode of communication but the underlying hypothesis behind grounded communication. Sec 4.4 in the paper talks about these findings without comparing the findings with what was found in Eccles et al.
> >
> > The other major concern related to the evaluation of compositional generalization in emergent languages is not addressed.

---

### Author Response · Authors · 2021-11-17
**Response to all reviewers**

We would like to thank all the reviewers for their constructive feedback and positivity towards our research direction. We especially appreciate the pointers towards related literature that we missed, along with improvements regarding the experiments.

To summarize the main changes in our updated submission:

1. We updated references to literature to include all the work that the reviewers pointed out.
2. Three reviewers pointed out that we did not have experiments that quantify compositionality. In the revised Section 4.3, we have added quantitative results using the metrics from (Brighton and Kirby, 2006; Chaabouni et al. 2020). Our results are comparable to that of (Lazaridou et al. 2018).
3. We compare our results with communication over an acoustic channel to results without the acoustic channel, i.e. direct discrete communication, using the RAIL baseline from  (Foerster et al. 2016).

We also included all the minor recommendations.  We give full details in our responses to each individual reviewer.

Once again we’d like to thank all reviewers for their kind considerations. We feel the additional literature and experiments have improved the paper as a whole.  We hope that we have addressed all the reviewers’ concerns but welcome further discussion.

---

### Decision · Program_Chairs · 2022-01-20

**Decision:**

Reject

**Comment:**

While a lot of previous work on emergent communications studies discrete protocols, this work explores a continuous and audio-based channel for learning multi-agent communication. Reviewers have commented positively on the novelty of the topic. At the same time, there are a number of concerns raised with respect to experimental design and implementation (6auy) and general approach of the topic which, as reviewers t576 and 42Xh point,  doesn't really go deep into the analysis and understanding of the particular experimental setup and findings. So, unfortunately as the papers stands I cannot recommend acceptance at this time. However, given that continuous communication in emergent communication is a somewhat overlooked topic, I would encourage the authors to use the reviewers' feedback and strengthen their manuscript.